# Interfacial Dynamics of Adsorption Layers as Supports for Biomedical Research and Diagnostics

Eva Santini [1,*], Irene Nepita [1,2], Alexey G. Bykov [3], Francesca Ravera [1], Libero Liggieri [1], Saeid Dowlati [4], Aliyar Javadi [5], Reinhard Miller [6] and Giuseppe Loglio [1]

1   CNR-ICMATE—Institute of Condensed Matter Chemistry and Technologies for Energy, 16149 Genova, Italy
2   Nanoscopy & NIC@IIT, Istituto Italiano di Tecnologia, 16163 Genoa, Italy
3   Institute of Chemistry, St. Petersburg State University, 198504 St. Petersburg, Russia
4   Chair of Chemical and Process Engineering, Technical University of Berlin, 13355 Berlin, Germany
5   Institute of Radiopharmaceutical Cancer Research, Helmholtz-Zentrum Dresden-Rossendorf, 01328 Dresden, Germany
6   Institute for Condensed Matter Physics, Technical University Darmstadt, 64289 Darmstadt, Germany
*   Correspondence: eva.santini@ge.icmate.cnr.it

**Abstract:** The input of chemical and physical sciences to life sciences is increasingly important. Surface science as a complex multidisciplinary research area provides many relevant practical tools to support research in medicine. The tensiometry and surface rheology of human biological liquids as diagnostic tools have been very successfully applied. Additionally, for the characterization of pulmonary surfactants, this methodology is essential to deepen the insights into the functionality of the lungs and for the most efficient administration of certain drugs. Problems in ophthalmology can be addressed using surface science methods, such as the stability of the wetting films and the development of artificial tears. The serious problem of obesity is fast-developing in many industrial countries and must be better understood, while therapies for its treatment must also be developed. Finally, the application of fullerenes as a suitable system for detecting cancer in humans is discussed.

**Keywords:** dynamic surface tension; dynamic interfacial tension; dilational surface visco-elasticity; drop profile analysis tensiometry; bubble pressure tensiometry; Langmuir trough; Brewster angle microscopy; medical applications



## 1. Introduction

At the present time, natural and life sciences are moving closer and closer together. Although the "fundamental" parts of mathematics could still exist independently and separately from other sciences, their real importance for other scientific branches is their impact, most of all on physical and chemical sciences. These natural sciences, in turn, are becoming more and more involved in life sciences, meaning that an efficient multidisciplinary system has resulted, in which the various branches stimulate and also require each other's inputs. The fundamental principles of surface and interfacial sciences are wide-spread in nature and technology, and medicine as important part of the multidisciplinary field profits a lot from the intensified exchange of knowledge. Much progress in modern medicine is often caused by the application of specific chemical and physical principles, and in particular colloid and surface sciences provide many tools to better understand biological systems [1].

This article deals with only a few examples from selected subjects to demonstrate how experimental instrumentation, theoretical approaches, and model studies allow insight to be gained into very complex situations. For a detailed description with more examples, even the space of a whole book would be insufficient. The first topic deals with the application of dynamic surface tensiometry of human body liquids to support medical diagnostics and to evaluate the success of therapies for various diseases [2]. In fact, this technique can be considered a useful tool for complementing other long-established methods.

A second group of applications of surface science principles is discussed with respect to the formulation of artificial tears and of synthetic lung surfactants. Understanding the mechanisms of these biological liquids and their design and application is vital for the successful treatment of malfunctions of the eye [3] or of the lungs, respectively. In particular, due to the still-existing COVID-19 pandemic, it was a hot topic to show in a Special Issue of a journal, summarizing various aspects of surface science applications in pulmonology, how essential the understanding of the mechanisms of pulmonary surfactants in the human lungs is [4,5].

A particular issue in industrial countries is the large number of people suffering from obesity. Surface science methods based on single-drop tensiometry with real-time drop volume exchange have mimicked the rather complex process of digestion in the human body. This successful work helped to develop strategies for the control of energy intake, and in this way have contributed to the serious problems caused by overweight [6].

The final example discussed here is dedicated to the use of fullerenes for the diagnosis of various types of cancer. Noskov and his team were able to demonstrate how surface science experiments on the monolayer properties of various fullerenes are essential in combination with other methodologies to detect the respective diseases [7].

## 2. Dynamic Surface Tension and Dilational Surface Visco-Elasticity Methods as Tools for Diagnosis and Therapy Control in Medicine

### 2.1. History of Tensiometry Studies in Medical Research

The human body contains many biological liquids, such as blood, serum, urine, seminal plasma, amniotic fluid, saliva, cerebrospinal fluid, tears, mammalian milk, sweat, lymph, and gastric fluid. The compositions of these liquids are very different, and depends on the functionality, age, and gender of the person. However, they all contain proteins, low molecular weight surface-active compounds, and salts. From fundamental studies of aqueous solutions of proteins, surfactants, and their mixtures, we know that the time dependence of the surface tension and the frequency dependence of the corresponding surface dilational visco-elasticity are strongly correlated with the type, concentration, and composition of the respective solutions [8]. Hence, the dynamic surface tension and dilational visco-elasticity of the various human liquids should be very different, and it can be expected that they change with the health state of a person.

Looking into the history, Polányi was probably the first to study the surface tension of human biological liquids [9]. He measured the surface tension of cerebrospinal liquids. Later, Künzel in 1941 worked on this topic and performed the first systematic analysis of dynamic surface tension data of blood serum and cerebrospinal liquid [10]. He showed that first of all the cerebrospinal liquid contains surface-active compounds that adsorb and decrease the surface tension. Moreover, he stated that the blood serum contains even more of these surface-active molecules so that the corresponding surface tension value is lower.

After these historical studies, no real systematic investigations of human biological liquids with respect to their physico-chemical surface properties happened for quite some time. The real activity on this topic started from the end of the last century, when people studied the surface tension behavior of various liquids [11], such as of blood serum [12,13], cerebrospinal [14] and amniotic liquids [15], gastric juice [16,17], synovial liquid [18], and saliva [19–21].

In one of the medical branches, pulmonary medicine, and here in particular neonatology, measurements of surface tension used for studying the functionality of pulmonary surfactants have become very popular. Remarkable progress has been made using this surface science tool for the treatment of pulmonary diseases, as summarized by Pison et al. [22]. Researchers have used expired air condensates [23] and also artificial pulmonary surfactants [24,25]. Not only are dynamic surface tension measurements suitable to help improve the diagnostics of lung diseases, but in particular investigations of thin liquid films stabilized by natural or artificial pulmonary surfactants are also able to provide essential information about their suitability for the efficient functioning of the lungs [26].

The interfacial characterization of biological liquids using different tensiometric techniques has deepened the knowledge of their properties, both in well-functioning organs and in critical conditions. Hence, the physico-chemical investigations may support the formulation of synthetic surfactants able to be employed for the treatment of different kinds of diseases.

### 2.2. The Standard State of Tensiometry Data for Human Liquids

In order to use the surface characteristics of human liquids, a base line must be determined in order to know what the normal values are and how they may depend on gender and age [2]. Only then can experiments be performed in order to find out if particular diseases cause significant changes in one or more characteristic surface parameters. Figure 1 shows the dynamic surface tension values of blood sera taken from newborns during their first day of life. Curve 1 relates to a well-functioning lung, while curves 2 and 3 indicate that the lungs of these babies are not sufficiently stabilized for normal breathing. Hence, the use of dynamic surface tension data could easily serve as a quick test of the maturity of a newborn, similar to what was proposed by Nikolov and Exerowa via the study of the stabilization of a liquid film by amniotic fluid taken before the birth of the baby [27].

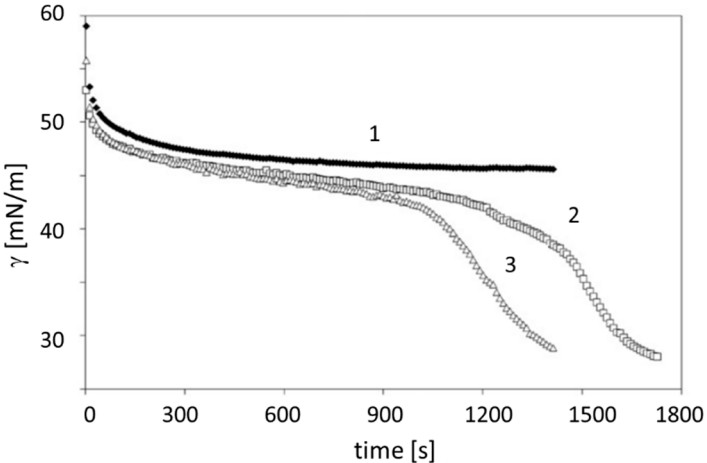

**Figure 1.** The dependence of the dynamic surface tension of the umbilical blood taken during the first day of life: (1) premature newborn from the reference group; (2,3) two newborns with immature lungs suffering from congenital pneumonia; data taken with permission from [28].

It has been demonstrated that the gestation period and the dimensions of the fetus are directly related with the concentrations of specific lipids and proteins (e.g., cholesterol, triglycerides, and atherogenic). During the pregnancy, the protein and lipid compositions vary with time, consequently affecting the surface tension of biological liquids. This assumption is supported by experimental data, as depicted in Figure 2, where the development of some selected dynamic surface tension values is shown during the pregnancy. Note that the data were not obtained from a single patient but are averaged data for many studied patients (52 women). Here, $\gamma 1$, $\gamma 2$, and $\gamma 3$ are the surface tension values for three different time periods of the adsorption kinetics, namely t = 0.01 s, 1 s, and at equilibrium (t→∞), respectively. These three selected dynamic surface tensions are most informative and can be used for the evaluation of the health state of patients suffering from different diseases. Using the bubble pressure combined with bubble profile analysis tensiometry, these data are easily accessible [8]. Such data are not only used in human medicine, but also in veterinary medicine, as impressively shown in [29].

It becomes clear that the respective circumstances of a patient, such as pregnancy, can lead to significant changes in the dynamic surface tension behavior of their blood serum.

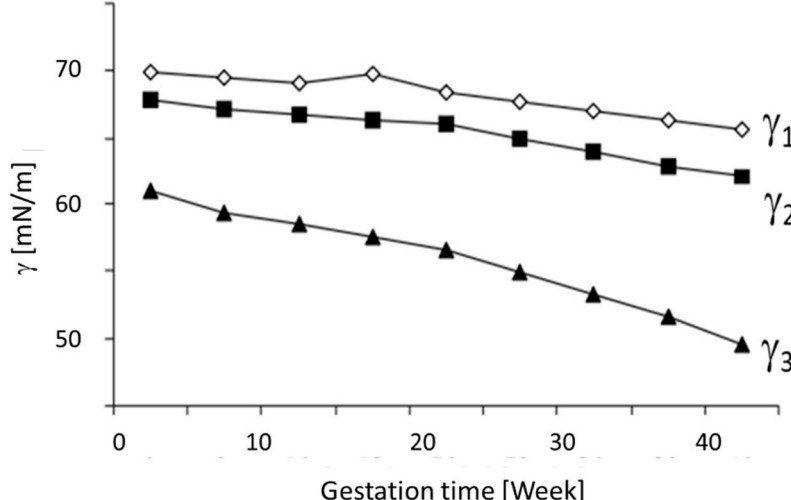

**Figure 2.** Changes in selected surface tension parameters for blood serum during pregnancy. Data taken with permission from [2].

### 2.3. Dilational Surface Visco-Elasticity Data for Human Liquids

From a surface science point of view, the measurement of the dynamic surface tension is not the only accessible technique for gathering essential data that are helpful in diagnosis and therapy control in human medicine. In 2008, Kazakov et al. [30] analyzed the surface visco-elasticity of composed protein–surfactant mixtures as well as real human liquids using an oscillating drop methodology. It was possible to demonstrate that in particular situations, the dilational visco-elasticity can provide more specific data for the diagnostics of certain diseases as compared to the dynamic surface tension data. For example, in Figure 3, the visco-elasticity modulus $|E|$ and phase angle $\phi$ of the blood serum before and after an operation is shown, and their significant change in terms of their frequency dependence demonstrate the medical potential of these easily accessible surface parameters.

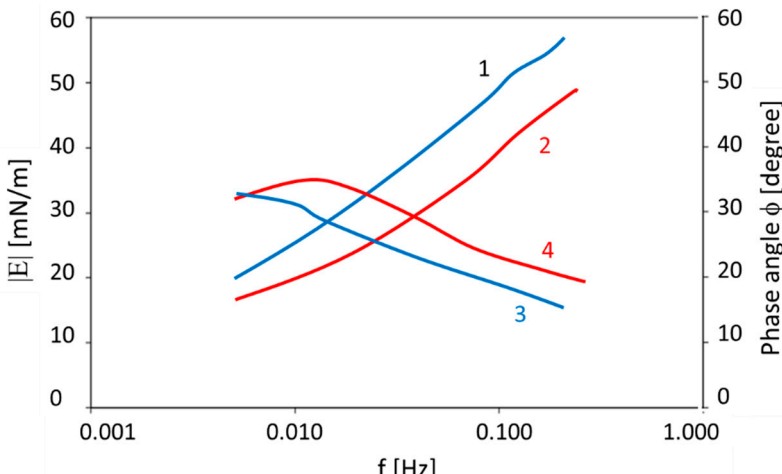

**Figure 3.** The visco-elastic modulus $|E|$ (1, 2) and phase angle (3, 4) values as a function of the oscillation frequency $f$ measured for blood serum: before the operation (2, 4) and after the re-canalization of the biliary tract (1, 3); data taken with permission from [30].

Many more examples of the information obtained from the dilational visco-elasticity of blood serum, urine, breath condensate, and cerebrospinal liquid samples can be found. Such investigations are essential for several medical fields, such as rheumatology, neurology, and pulmonology [31]. In particular for the characterization of pulmonary surfactants and

their synthetic substitutes, surface relaxation methods are very efficient, as recently shown in [32] and further discussed below in more detail in Section 5.

In fact, evident progress has been achieved thanks to the interfacial properties studied in pulmonary medicine, and specifically for lung surfactants related to the neonatology field. This is expected to lead to similar results for other branches of medicine via the systematic and standardized utilization of the tensiometric methodologies. From this perspective, the investigation of surfactant systems by simulating real human liquids under the influence of interfacial properties is fundamental for using this approach as a complementary method to other medical techniques, as discussed in the following sections.

## 3. Artificial Tears for Dry Eye Syndrome

According to the international Tear Film and Ocular Surface Society (TFOS), dysfunctional tear syndrome (DTS), commonly known as "dry eye syndrome", is defined as an "ocular surface disease characterized by a loss of homeostasis of the tear film, and accompanied by ocular signs, in which tear film instability and hyperosmolarity, ocular surface inflammation and damage, and neurosensory abnormalities play etiological roles" [33,34]. DTS is a common ocular disorder affecting millions of people, which causes decreases in both the quality and amount of tears [35]. Human tears have the vital function of moisturizing the ocular surface and minimizing damage to the corneal epithelium. The tears are mostly composed of water, electrolytes, proteins (e.g., antibodies and lysozymes), and lipids, which combined together form three distinct layers (see Figure 4):

(1)     The outermost lipid layer produced by the meibomian glands is located in the eyelids and composed of both polar and non-polar lipids,
(2)     A middle aqueous layer produced by the lacrimal glands,
(3)     The epithelium-covering mucoid layer, which helps to anchor the aqueous part of the tear film on the ocular surface.

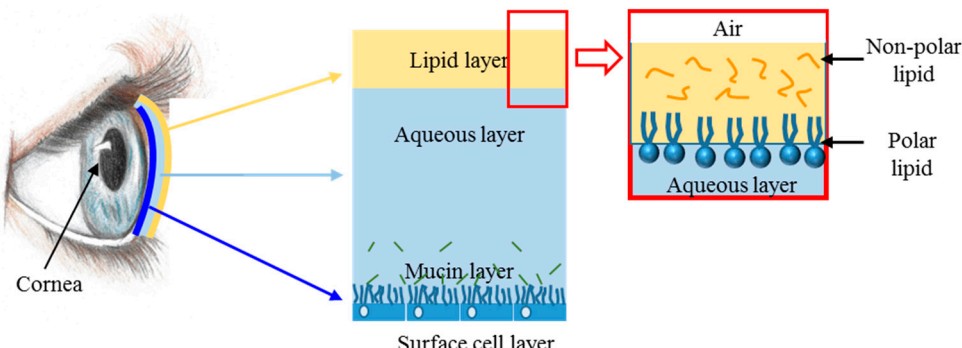

**Figure 4.** A scheme of the eye's lipid layers.

Dysfunctions in any of these layers can provoke tear film instability and hyperosmolarity [36] due to environmental factors, systemic diseases, and medications [37]. This kind of condition can lead to a wide variety of symptoms such as redness, stringy mucous, burning, and itchiness, eventually causing scarring and fibrosis due to unprotected corneal epithelial exposure [38]. There are two primary categories of DTS, aqueous deficient and evaporative [39]. Left untreated, both DTS categories can result in visual disturbances and tear film instability, with potential damage to the ocular surface [40], also increasing the risk of ocular infection. Tear substitutes are necessary for the treatment of these kinds of ocular disease.

Tear substitutes have been classified by Barabino et al. [41] into three categories, depending on their degree of interaction with the eye: wetting agents, multiple-action tear substitutes, and ocular surface modulators. While wetting agents only lubricate the ocular surface with a limited residence time, multiple-action tear substitutes can improve the tear film quality and quantity without interacting with the ocular surface. The last category, ocular surface modulators, interacts with the ocular surface in order to counteract DTS.

Artificial tears are the preferred first-line therapy as tear substitutes due to their non-invasive nature and low side effect profile, despite the fact that the currently commercial products do not successfully recreate the normal human tear film, although they mimic its behavior well. A vast number of artificial tear products are currently available on the market, in multidose vials containing preservatives or in preservative-free single-dose units, which are used to assess a formulation's sterility and avoid eye infections. The most common kinds of ingredients contained in artificial tears can be summarized as follows [33]:

(1) Viscosity-enhancing agents are used to increase the tear film thickness and the retention of artificial tears at the ocular surface [42], preventing the loss of water, since they act as water-retaining agents;

(2) Electrolytes are able to maintain the osmotic balance of the ocular surface by providing essential ions for the maintenance of the corneal epithelial cells [43];

(3) Osmoprotectants are utilized to prevent ocular surface cell apoptosis induced by DTS.

(4) Regarding oily agents and surfactants, the presence of lipids and proteins in the lipid layer plays a critical role in the surface tension of the tear film and humectation of the ocular surface. Any alterations in the lipid layer lead to an increase in tear evaporation [44]. Consequently, oily agents, in the form of liposomes and nanodroplets, are used in the formulations of tear substitutes to replenish this layer [45].

There is a need to better understand the specific mechanical and pharmacological roles of each ingredient composing the different formulations in order to establish the behavior of the lipid layer, and in particular, its interfacial properties, such as the interfacial tension and visco-elasticity.

In order to mimic a realistic layer of meibum secretion, a model system composed of cholesteryl esters and cholesterol can be investigated, as they constitute the major components of the lipid layer. The studies described in [46] were carried out using the Langmuir trough technique for characterizing both the equilibrium and dynamic interfacial properties of the cholesterol (CH) and cholesteryl stearate (CS) monolayers. In fact, it is convenient to measure the dilational surface moduli of elasticity and viscosity under dynamic conditions, since the meibum undergoes numerous mechanical perturbations from its formation until its renewal upon blinking. Even if the real disturbance of the tear fluid does not respond to a periodic sinusoidal deformation, measurements at frequencies close to 0.1 Hz have been performed, taking into consideration that a typical blinking period is around 5 s [47]. On the other hand, the evaluation of the equilibrium properties of the lipid monolayers is also necessary to gain a complete picture of their dynamic behavior. In fact, an analysis of the surface pressure area per molecule isotherms has shown the existence of a strong synergy between cholesterol and cholesteryl stearate. The addiction of cholesterol to the lipid mixture increases the rigidity of the monolayer, with a consequent enhancement of the dynamic surface visco-elastic modulus. It has been shown that the addition of CH may induce conformational modifications, leading to the existence of different types of organization for the molecules at the interface:

(a) At the lowest surface pressures a disordered liquid phase is observed, due to a weak lateral packing of the molecules;

(b) By increasing the surface pressure, the coexistence of the two liquid phases is expected, with a decrease in elasticity;

(c) A further increase in elasticity then occurs due to an enhanced lateral packing of the molecules.

Figure 5 shows the dilational elasticity as a function of the molar fraction of cholesterol (xCH) for experiments performed at two different deformation frequencies around three different values of the reference pressure, at the physiological temperature of 35 °C. As demonstrated, the elasticity modulus is almost unaffected by the deformation frequency, and the decrease in dilational surface elasticity indicates the formation of more fluid monolayers.

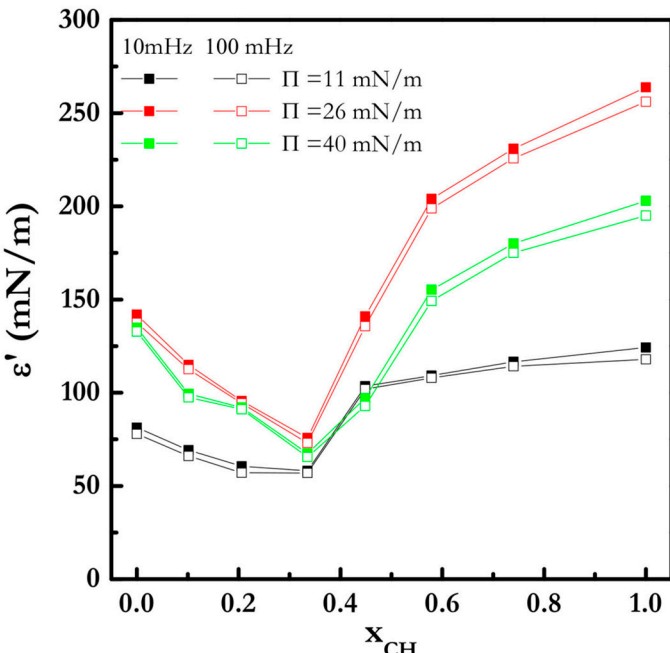

**Figure 5.** The surface elasticity moduli of the CH/CS mixture vs. xCH at different surface pressures at 35 °C; taken from [46].

The study of this simple model system has shown that slight modifications of the composition of the lipid monolayers may induce strong changes in the interfacial properties, which in turn may cause physiological dysfunctions of the meibum film, and in turn a DTS pathology. For this reason, it is necessary to define the best composition for mimicking the biological characteristics of the meibum layers and deepening the aspects related to different ocular diseases [48].

The study of the dynamic and equilibrium surface properties is also fundamental for determining the optimal composition and the behavior of another class of tear substitutes: oil-in-water emulsions, where oily droplets are stabilized in the aqueous phase using suitable emulsifiers. As previously mentioned, the lipid layer is naturally composed of both non-polar and polar lipids, and it has been determined that polar lipid abnormalities may be involved in DTS [49,50]. Therefore, several formulations include polar lipid-like surfactants, since in cationic emulsions the positively charged oil droplets can interact with the negatively charged mucin layer of the tear film to stabilize it [51,52]. In these cases, studies of the interfacial properties are essential for the choice of the suitable type and concentration of surfactants in a way to avoid toxic effects and increase the eye moisture. Moreover, the behavior of the surface tension and visco-elasticity provides important information for the formulation of stable oily emulsions [53] that are suitable to interact with and protect the lipid eye layer.

The determination of the physico-chemical effects of the tear substitutes on the lipid layer may be considered a complementary investigation in support of the physicians' activities for the choice of the most suitable tear substitute, depending on the specific ocular disease affecting each patient.

## 4. Effects of Serum Proteins on Interfacial Properties of Ophthalmic Silicon Oils

Vitrectomy is the second most frequently used surgical ophthalmic intervention after cataract, consisting of the replacement of the vitreous body, a visco-elastic fluid that occupies the vitreous chamber of the eye, with transparent vitreous substitutes. It is performed to treat several pathological or traumatic vitreoretinal conditions, such as retinal detachments, hemorrhages, inflammations and infections, and the presence of foreign bodies in the eye. After the vitreous removal, a long-term substitute is commonly injected to fill the vitreous chamber and the sensory layer of the retina in contact with the retinal pigment epithelium.

Despite the wide use of these vitreous substitutes and their relevance in ophthalmic surgery, none of the currently available products possess all of the physiological requirements for an efficient vitrectomy procedure, since they present significant shortcomings, mostly related to lack of biocompatibility and inadequate behavior [54].

High-viscosity (up to 2000 cSt) polydimethylsiloxane (PDMS) silicone oils (SOs) are among the most frequently used vitreous substitutes. To date, they remain an indispensable tool in retinal surgery, especially in complicated retinal pathologies requiring a long-term vitreous substitute for internal tamponade. Most SOs used in ophthalmic surgery have a density equal to 0.97 g/cm$^3$; hence, they float above the residual eye cavity fluid, which helps in retinal reattachment in the case of superior breaks. Heavier SOs with densities between 1.02 and 1.06 g/cm$^3$ are also used for other purposes. One of the important advantages of using SOs is related to their high surface energy at the interface with the aqueous phase (i.e., interfacial tension), which ensures the closure of retinal breaks and reduces subretinal leakage.

The use of SOs, however, is associated with some complications. There is clinical evidence [54] that if an SO is left for a longer period in the vitreous chamber, it invariably tends to emulsify. Besides the obvious, although reversible, affected vision caused by the light scattering, the occurrence of the emulsification leads to a series of possible serious complications requiring the early removal of the SO [55]. This is particularly true when the oil droplets are small enough to pass from the vitreous to the anterior chamber. In this case, the droplets may get trapped in the trabecular meshwork and block the aqueous drainage, leading to increased intraocular pressure and a higher risk of developing glaucoma.

Several experimental studies have proven the relevance of the emulsification of the exposition of SO in the vitreous chamber to endogenous molecules (proteins, lipids, etc.), the presence of which is favored by the post-surgery inflammatory state of ocular tissues. Bartov et al. [56] demonstrated that various blood constituents, such as lymphocytes, plasma, red blood cells, and hemoglobin, act as emulsifiers for SO when dissolved in aqueous solution. Heidenkummer et al. [57] characterized various SOs with specific physicochemical properties in terms of their rate of emulsification. The authors added biomolecules to the system, and in particular used 0.1% solutions of fibrinogen, fibrin, $\gamma$-globulins, acidic alpha-l-glycoprotein, and serum dissolved in sterile distilled water, as well as in balanced salt solution [57]. They found that the group of low-viscosity SOs (1000 cs) was the least stable. They identified as the most effective emulsifiers fibrinogen, fibrin, and serum, followed by $\gamma$-globulins. Emulsions obtained by sonicating and centrifuging mixtures of SO and saline solutions containing various blood components were investigated by Savion et al. [58], who found that red blood cell membranes, plasma lipoproteins, and purified (high-density lipoprotein) HDL apolipoproteins favored SO emulsification.

It is, therefore, evident that most of the above-mentioned biomolecules are surface-active molecules that adsorb at the SO-aqueous interface and favor its emulsification, based on their effects on interfacial properties. For comprehension of the emulsion's generation and stability, it is of great relevance [59–61] to understand how such molecules modify these interfacial properties, and in particular the dilational surface rheology; that is, the dynamic response of the interfacial tension (IFT) to perturbations of the interfacial area.

These effects have, however, been investigated only in a few studies. The IFT values between SO and pure water are in the range between 35 and 42 mN/m, depending on the specific SO composition [62,63]. In the presence of endogenous molecules, such as in a vitreous chamber, the equilibrium IFT of SO against the aqueous phase is significantly reduced. In fact, Nakamura et al. [64] reported an IFT value, measured at 37 °C using the Du Nouy ring method, of about 16 mN/m between SO (1000 cSt) and vitreous liquefied bovine fluid. In addition, they found similar values for the IFT of SO against intraocular fluids, and slightly lower values (12.6 mN/m) for SO in contact with retinal tissue specimens after 24 h. All of these IFT values were significantly lower than those measured for the SO–pure water interface and are compatible with an increased tendency for the system to emulsify.

More recently, in [65], the equilibrium and rheological properties were characterized for the interface between 1000 cSt SO and aqueous solutions (Dulbecco's saline buffer) containing albumin, γ-globulin, and their mixtures. These proteins represent the most abundant fractions of human serum, with 35–50 g/L for albumin and 20–55 g/L of globulins. The measurements were performed using pendant drop tensiometry. As for Figure 6, the results showed that for a protein concentration of the order of 20% of the physiological values, the IFT decreases down to values of about half of the value measured for the buffer–SO interface, with the protein mixture being slightly more surface-active. Measurements performed with oscillating drops [66] after the IFT equilibrium showed significantly larger values (see Figure 7) of the surface dilational visco-elasticity module for γ-globulins at concentrations that were only a few percent of the physiological value, over a broad range of oscillation frequencies. These values are, therefore, compatible with emulsions showing significant stability against droplet coalescence. Emulsification tests in syringes [67] on the same system demonstrated that while the SO–buffer interface is unable to provide an emulsion, in the presence of the above proteins, stable emulsions (over several months) are obtained already at concentrations above 0.5% of the physiological concentration in blood and even of the order of 0.01% for protein mixtures. This finding corroborates the hypothesis that the release of a small amount of proteins from ocular tissues (typically as an inflammatory response to surgery) may have a primary role in the formation of an emulsion.

Considering the relatively low IFT values, emulsification could be caused by typical head movements, or more likely by voluntary or involuntary eye movements (saccadic movements of the eye, eye rotation during sleep). The latter could also promote the interfacial area perturbations related to dilatational stimuli of the interfacial layer.

In addition to biomolecules, other types of molecules can adsorb and modify the interfacial properties of SO. Despite the use of purification and ultra-purification processes, variable contents of low molecular weight components (LMWCs) can be still detected in purified SOs, and these compounds are known to act as surfactants [68]. Moreover, comparing solutions of buffer, human serum albumin (HSA), and SOs of different compositions, the SOs with larger concentrations of LMW silicones have been associated with increased protein denaturation and aggregation and SO-in-water emulsions [67]. Dresp and Menz [69] investigated the effect of detergent contamination on a ready-to-use standard set of vitrectomy instruments. They concluded that for reusable instruments, the remnants of cleaning substances from the sterilization process can increase the risk of the emulsification of the SO due to a significant decrease in the respective IT value.

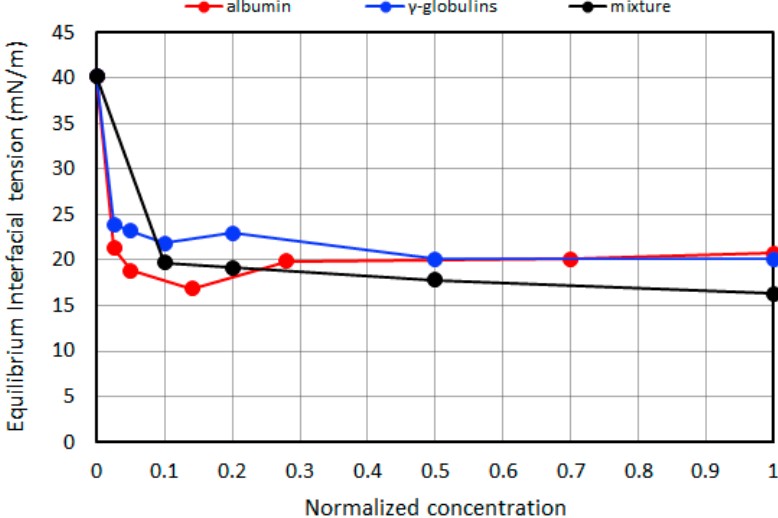

**Figure 6.** The equilibrium IFT as a function of the protein concentration, expressed as a fraction of the physiological content in the blood; adapted from Nepita et al. [65].

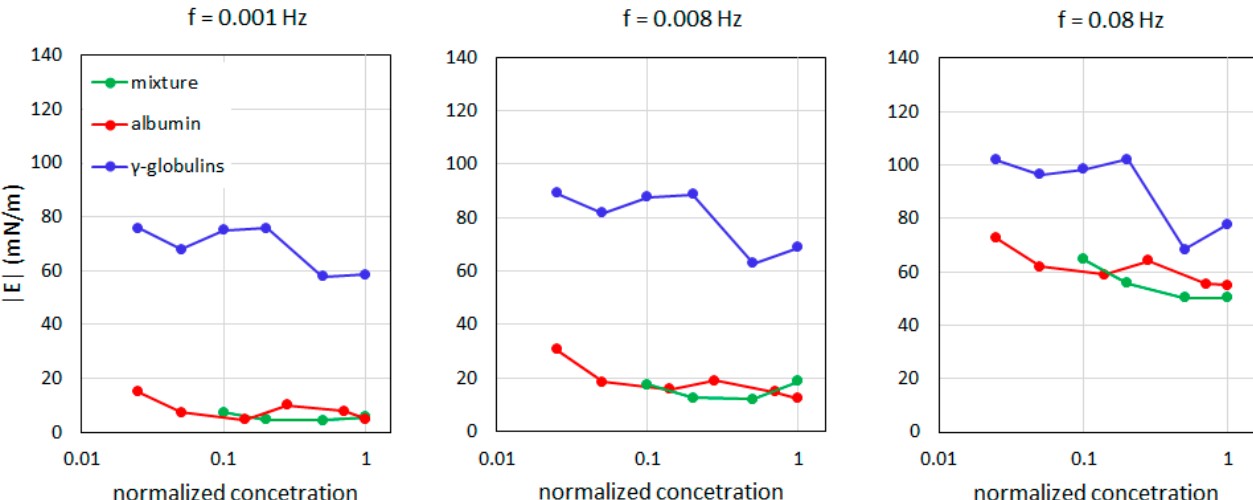

**Figure 7.** The surface dilational visco-elasticity modulus |E| values as a function of the protein concentration, expressed as a fraction of the physiological content in the blood according to Nepita et al. [3].

## 5. Native and Synthetic Pulmonary Surfactants for Neonatal Respiratory Distress Syndrome

### 5.1. Native Pulmonary Surfactants

Thin films of pulmonary surfactant solutions cover the inner surfaces of the lungs and play an important role in the process of breathing [70]. A pulmonary surfactant is a complex mixture of lipids with proteins and its deficiency due to prematurity causes respiratory distress syndrome, which is the leading reason for premature infant death [71]. The pulmonary surfactant decreases the surface tension to extremely low values after surface compression and prevents alveolus collapse upon exhalation [72]. This unique ability to decrease the surface tension to almost zero arises due to a significant amount of dipalmitoyl phosphatidyl choline (DPPC), with its molecular structure helping to form a closely packed monolayer [70–72]. On the other hand, this molecular structure prohibits DPPC adsorption from the bulk solution, where the lipid remains in the form of vesicles [73]. Initially, the surface tension values at equilibrium and after compression were the main parameters used to characterize pulmonary surfactant solutions [74]. Only the kinetic dependencies of the surface tension at adsorption of the pulmonary surfactant on pure aqueous surfaces were measured to estimate the dynamic surface properties [75]. This allowed the possibility of proposing a medical treatment procedure for premature infants based on natural pulmonary surfactants extracted from animal lungs [71]. At the present time, several pharmaceutical formulations are used (Curosurf, Infasurf, Survanta, Alveofact) that save thousands of lives per year, but like many other natural medical drugs, they have several disadvantages, including their unstable composition, high costs, and limited sources [76]. The first generation of synthetic pharmaceutical formulations of pulmonary surfactants (ALEC and Exosurf) was made based on DPPC and had low efficiency, although they met the requirements of the parameters presented previously [77]. This means that these parameters proved to be insufficient for the full characterization of the inner surfaces of the lungs, since the real state of the lungs proved to be more complicated. For example, alveoli undergo continuous deformation and remain far from equilibrium, while the minimal values of surface tension strongly depend on the magnitude of deformation and adsorption from the bulk of pulmonary surfactants that usually proceeds on the preoccupied surface.

### 5.2. Dynamic Surface Properties of Pulmonary Surfactants

Investigations of surface properties of pulmonary surfactant solutions via surface rheology and other experimental techniques have given additional information on the key components [24,78,79]. It was shown that pulmonary proteins, especially SP-B, facilitate

the formation of the multilayer structure in the adsorption layer, which serves as reservoir for supplementary molecules staying close to the surface [3,80]. The slow progress in this scientific area can be explained by the significant difficulties connected with the measurements of dynamic surface properties at conditions relevant to the physiological state inside the lungs, with low surface tension values (lower than 30 mN/m). Several experimental methods were developed for measuring extremely low surface tension values at periodic compression and expansion levels [81]. One of the frequently used methods involves a captive bubble surfactometer, which is based on an analysis of bubble shapes in solutions of pulmonary surfactants. This allows the simulation of tidal breathing by changing the bubble volume. Recently, the constrained drop surfactometer method was developed, where the drop of a pulmonary surfactant solution is used instead of a bubble. In this case, the composition of the air phase is under control, which makes it possible to investigate the influence of foreign particles or agents on the surface properties of pulmonary surfactant solutions [82]. The Langmuir film balance with a special design of the Langmuir trough for work in the range of low surface tension is also frequently used for the investigation of different model systems, and can be easily combined with optical or spectroscopic techniques. It works at relatively slow deformation speeds with frequencies of the surface area oscillations of less than 0.2 Hz, but it allows the compression to be continued even after the surface tension reached zero values. The character of the surface tension changes as a response to deformations of the surface layer, directly connecting it with the functionality of the pulmonary surfactant [83,84]. The dynamic surface elasticity is a fundamental property of a surface layer describing the system response to deformations. It could be used to estimate the efficiency of pulmonary surfactant solutions. However, long-term measurements of the dynamic surface elasticity were performed only at sufficiently high surface tensions [24]. For measurements of the efficient surface elasticity in the range of low surface tension values, a new approach was recently proposed based on the analysis of the non-linear system response to large deformations [85]. It was shown that the surface tension changes during periodic compression and expansion phases, meaning the spread of the DPPC monolayer differs significantly from the corresponding changes for the adsorption layer of the native pulmonary surfactant (Figure 8) [32,86].

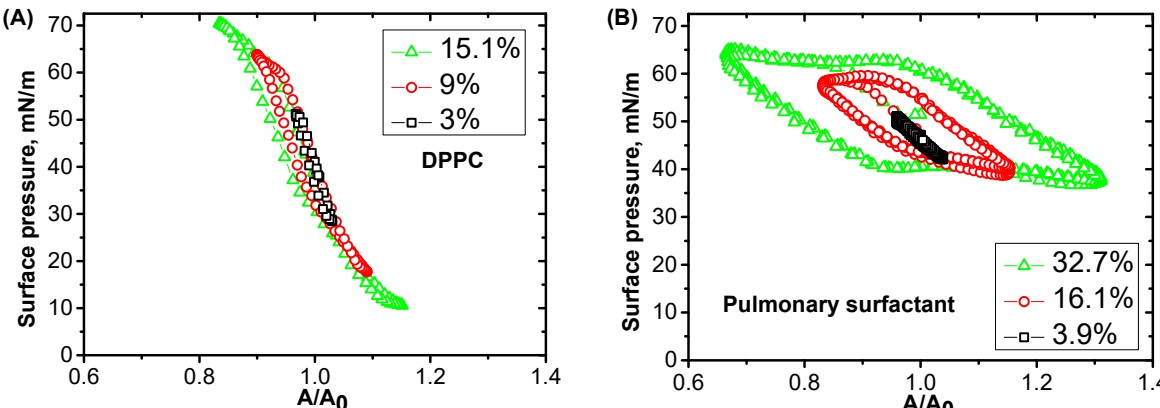

**Figure 8.** The dependencies of the surface pressure on the relative deformation A/A0 (Lissajous plots) measured at different amplitudes of the area oscillations, as indicated in the figure, for a spread monolayer of DPPC (**A**) and adsorbed layers of pulmonary surfactant solution at a concentration of 1.25 mg/mL at 25 °C (**B**).

At conditions close to the physiological conditions, the surface elasticity proved to be higher for DPPC than for the complex mixture of the pulmonary surfactants (Figure 9). It was assumed that the presence of proteins during the formation of a multilayer structure leads to an acceleration of the relaxation processes connected with an exchange of lipid molecules between the surface and sublayer. The temperature increase results in an additional acceleration of the relaxation processes due to the decrease in ordering in the surface

layer structure. For pulmonary surfactant solutions at 35 °C, the characteristic relaxation time decreases from hundreds to a few seconds with the decrease in surface tension from 30 to 1 mN/m [86,87]. Moreover, for deformations larger than 10%, an irreversible collapse in the adsorption layer takes place and leads to a loss of functional properties for some molecules, which have to be substituted by new molecules arriving from the bulk phase. In this case, a sufficiently fast transfer of molecules from the bulk phase or sublayer is required to maintain the stationary oscillation of the surface tension. Recently, it has been shown that the complexes of protein SP-B with anionic lipids and SP-C not only accelerate the adsorption of lipids to the surface layer, but also improve the penetration of hydrophobic and hydrophilic molecules through the vesicle's membrane [32,88].

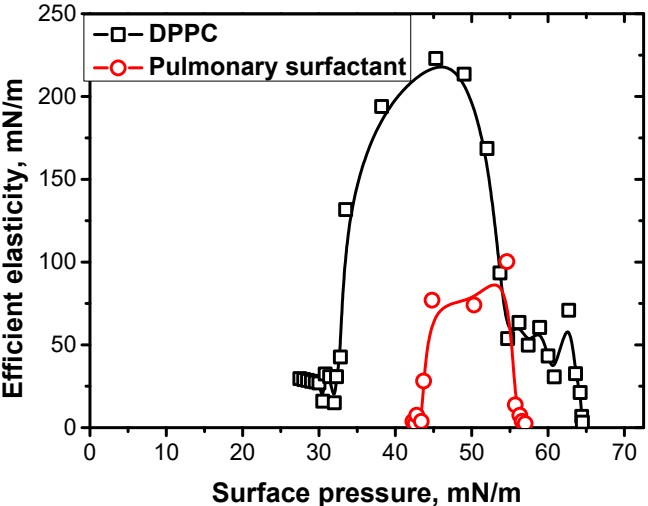

**Figure 9.** The dependencies of efficient surface elasticity at a frequency of 0.03 Hz and temperature of 35 °C on the surface pressure for a spread monolayer of DPPC (black squares) and adsorbed layers of pulmonary surfactant solutions at a concentration of 1.25 mg/mL (red circles). Data taken from [85,86].

### 5.3. Synthetic Pulmonary Surfactnts

In spite of the significant progress in understanding the proteins' functions in pulmonary surfactants, the strict mechanism of their actions, which allows the possibility for the pulmonary surfactants to maintain the functional properties of the lungs in the course of periodic surface deformations, remains unknown so far. Therefore, developing a second generation of synthetic pharmaceutical formulations for pulmonary surfactants is based on copying or imitating the molecular structure of SP-B proteins [76,77,84]. To date, only a part of the SP-B macromolecule can be synthesized, which will have to undergo medical trials [89]. At the same time, there are encouraging results for the influence of synthetic polyelectrolytes and co-polymers on the dynamic surface properties of pulmonary surfactants [90–92].

The applications of pulmonary surfactants for medical treatments of adult patients with acute respiratory distress syndromes have shown controversial results, probably due to differences in the administration methods. However, the COVID-19 pandemic has restored interest in and initiated several medical treatments for patients with this devastating disease. The preliminary results have shown that pulmonary surfactants can help patients on artificial lung ventilators [93–95]. Moreover, pulmonary surfactants are considered perspective agents for drug delivery to the lungs [96,97].

## 6. Interfacial Aspects of Digestion

### 6.1. Mechanism of Lipid Digestion

Lipase is an enzyme responsible for the digestion of lipids in our body. Triglycerides, also known as triacylglycerols, are the most common sources of lipids and the primary

substrates of lipase [98]. Triglycerides have a glycerol backbone attached to three fatty acid sidechains via ester bonds. Lipase can cleave these ester bonds during the digestion of lipids and produces diglycerides, monoglycerides, glycerol, and fatty acids [99]. Additionally, phospholipids, composed of a glycerol backbone attached to two fatty acids and a phosphate headgroup, can be digested using a specific group of lipases called phospholipases [100].

Human gastric and pancreatic lipases are the main enzymes for the digestion of lipids in the gastrointestinal tract, facilitating fat absorption by producing free fatty acids [101]. Since lipase is hydrophilic, the hydrophobic lipase substrates should be emulsified to maximize the enzymatic lipolytic digestion at the triglyceride–water interface [102]. The ingested lipids compose a coarse oil-in-water emulsion in the mouth, which becomes finer and finer along the stomach and small intestine. The competition at the interface of the emulsion droplets is the vital factor determining the rate of fat digestion. Besides the physical peristaltic forces, bile salts, phospholipids, lipolysis products, and proteins mediate the emulsification and determine the fate of the digestion [103].

### 6.2. Mimicking In Vitro Digestion

Owing to the moral and technical challenges of in vivo gastrointestinal studies, in vitro models are usually chosen to mimic digestion. To this end, interfacial techniques can be used to distinguish the bulk from interfacial phenomena [104]. Dynamic tensiometry coupled with subphase exchange fits when studying triglyceride–water interfaces similar to the passage of food colloids through the gastrointestinal tract by alternating the aqueous phase [105]. The OCTOPUS [106] and CDC-PAT [107] systems are two setups developed based on the droplet exchange technique. A schematic representation of an in vitro digestion model based on dynamic tensiometry is shown in Figure 10. During the formation of the initial protein adsorption layer, the dynamic surface tension is measured for 30–60 min. Then, interfacial oscillation is imposed on the droplet to measure the dilational rheology of the adsorbed layer. Afterward, the droplet bulk is exchanged with a new subphase, and the interfacial properties are measured for each of the exchanged solutions [106].

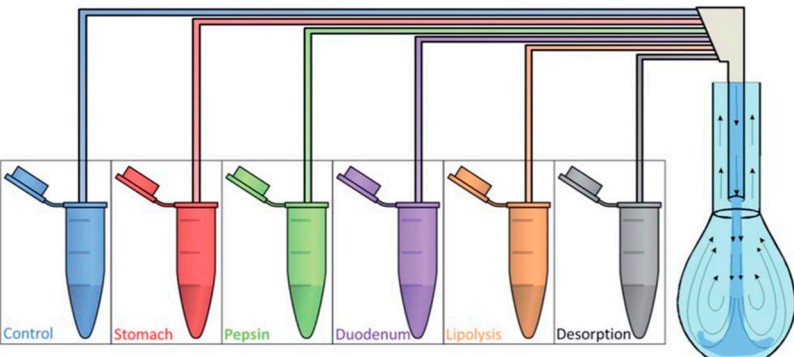

**Figure 10.** A schematic representation of the in vitro digestion procedure using the OCTOPUS setup based on a multiple-exchange protocol for the droplet subphase; reprinted with permission from [106].

### 6.3. Interfacial Dynamics of Digestive Oil–Water Interface

The surface-active molecules affect the enzymatic activity of the lipase by controlling its interfacial concentration [108]. Sn-2 monopalmitin and Sn-2 monocaprilyn, two monoglycerides produced during digestion, are highly surface-active and can desorb lipase from the interface, leading to a self-regulated rate of digestion [109,110]. Figure 11 shows how fast Sn-2 monoglycerides displace lipase from the interface. The surfactants can also interrupt the enzymatic activity of lipase as a model amphiphile. Cationic surfactants interact more strongly with lipase at the interface at neutral pH than anionic or non-ionic surfactants [111,112].

The effect of in vitro gastrointestinal digestion on protein-covered interfaces is shown in Figure 12. While the extent of lipolysis is similar for both proteins, the emulsification increases the pepsinolysis of β-lactoglobulin considerably, unlike the β-casein [106]. During pepsinolysis, an increase in the dynamic interfacial tension indicates the enzymatic degradation of proteins and their removal from the interface. During lipolysis, the interfacial tension decreases due to lipase adsorption and the generation of reaction products. The dilational visco-elastic moduli of the interface allow insight to be gained into the interfacial composition.

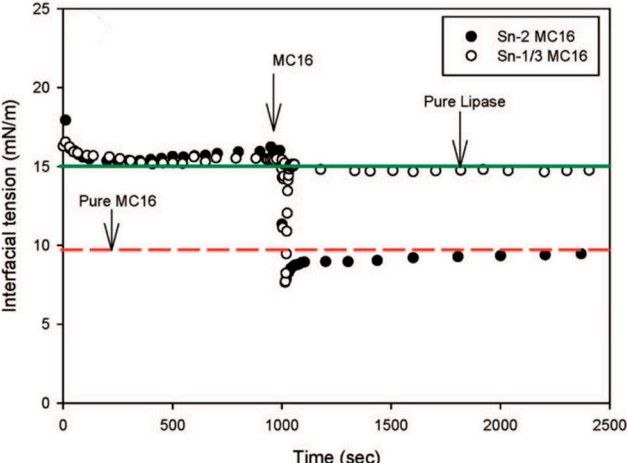

**Figure 11.** The interfacial tension of lipase-adsorbed layers at the buffer–decane interface followed by the injection of $6.7 \times 10^{-4}$ M Sn-2 monopalmitin and Sn-1/3 monopalmitin. The green and red lines show the equilibrium IFT for pure lipase and pure Sn-2 monopalmitin; reprinted with permission from [110].

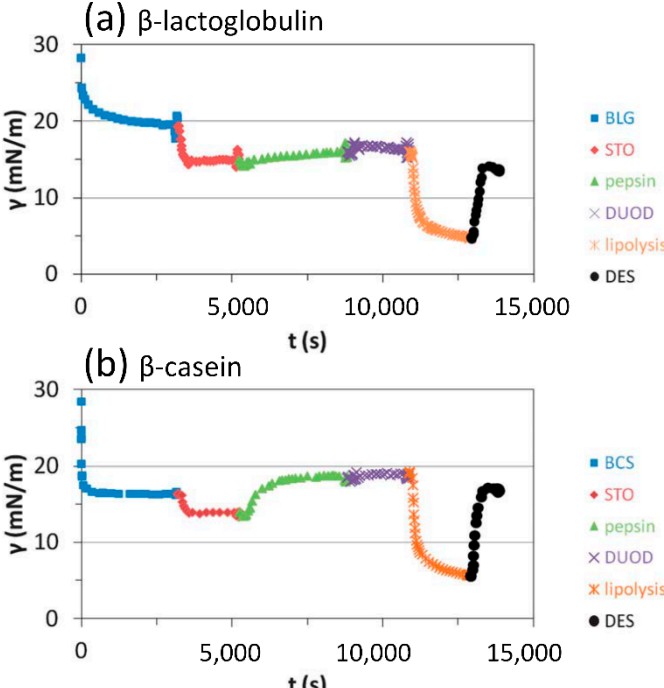

**Figure 12.** Mimicking the in vitro digestion at the olive oil–water interface with initially adsorbed layers of β-lactoglobulin (**a**) and β-casein (**b**); the gastric buffer (STO), pepsin, duodenal solution (DUOD), lipase and bile salt solution (lipolysis), and duodenal solution (DES) are exchanged into the droplet sequentially (reprinted with permission from [106]).

### 6.4. Bile Salt Effect on Lipase Activity

The lipolysis rate is determined by the competition and interactions of the surface-active materials at the interface and via the interfacial accessibility for lipase. Bile salts are highly surface-active biosurfactants, which can remove other materials from food emulsion interfaces, thereby arranging the interface for lipase–colipase complex adsorption [113]. Therefore, dynamic interfacial techniques have been used to study the interactions of food emulsion components with bile salts. Accordingly, a competitive sequential adsorption study showed that protein films have more resistance to desorption by bile salts than the non-ionic surfactant Tween 20 [114]. The complexation of bile salts at the interface determines their stability; sodium taurocholate adsorbs at the water–air interface more conveniently and irreversibly than sodium glycodeoxycholate [115].

### 6.5. Effects of Lipase Inhibitors as Antiobesity Drugs

The inhibition of pancreatic lipase regulates the amounts of free fatty acids and mono-glycerides released and subsequently absorbed by our body. Therefore, they provide an approach for discovering potential antiobesity drugs [116]. Orlistat, also commercially known as Xenical® or Alli®, is a well-known antiobesity drug approved by the US Food and Drug Administration. Its working principle is based on the competitive inhibition of human pancreatic and gastric lipases, i.e., it binds to the catalytic triad of lipase to restrict the availability of triglycerides [117]. However, regarding the side effects, safer potential substitutes have been actively sought, including natural lipase inhibitors [118]. The DrugBank database at the University of Alberta was investigated to find pancreatic lipase inhibitors via molecular docking and addressing silibinin(A) and glutathione–disulfide as potential inhibitors. Then, their effectiveness was confirmed by measuring the lipid digestion in a pendant drop model [6].

## 7. Fullerene Derivatives for Special Tumor Therapy

Fullerene was early discovered in 1985 by Kroto, Curl, and Smalley, the Nobel Prize winners in chemistry in 1996. In the subsequent years, the potential applications of this substance in different technological fields appeared to be rapidly evident by virtue of the chemical reactivity of fullerene's molecular structure.

The beneficial applications in medicine of a variety of fullerene derivatives, over the course of two decades after fullerene's discovery, were exhaustively pointed out by Da Ros [119]. Their photodynamic effects, formation of composites with biopolymers, interactions with proteins, and solubility in aqueous phases constitute the essential functionalities of fullerene derivatives in pharmacology innovations and the ensuing tumor therapies [120–124]. The successful oncological therapies of the body-injected product involve local accumulation onto malignant cells via an interaction with the cell membrane or penetration between the membrane layers as a self-arrangement, due to the peculiar alteration of the malignant cell structure. This locally targeted strategy demands the existence of two-dimensional forms of the adopted fullerene derivative. In this medical context, recently Noskov and his coworkers envisaged a greater chance of medical success resulting from a deeper understanding of the dynamic properties and interfacial rheology for adsorption layers of fullerene derivatives. A continuation of this series of experimental investigations, conducted by these authors, proved the significance of the designed formulations in the two-dimensional layered structure of the applied pharmaceutical fullerene products, instead of taking into account their sole molecular species [7,125–128].

For the first time, Noskov et al. [125] reported measurements of the dynamic surface tension and surface dilational elasticity modulus for fullerene (C60) derivatives with lysine and arginine amino acids. The measurement values, together with the concomitant Brewster angle microscopy images, allowed important information to be acquired about the mechanism of adsorption layer formation in such a complex system, as well as its structure and adsorption kinetics behavior. A distinct dynamic surface elasticity is exhibited by the

two studied fullerene derivatives. Figure 13 illustrates the time evolution of the surface tension and dilational surface elasticity modulus for an aqueous solution of C60-lysine.

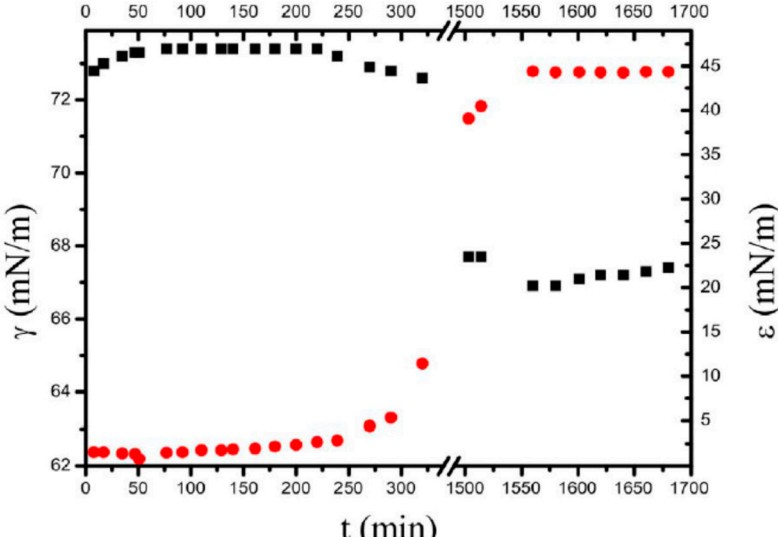

**Figure 13.** The kinetic dependencies of the surface tension (black squares) and dynamic surface elasticity (red circles) of C60-lysine solutions at the concentration of 0.5 g/dm$^3$; taken from [125].

Subsequently, Noskov et al. [126] studied the surface properties of solutions of fullerene derivatives with a high number of hydroxyl groups (denoted as fullerenols). The experimental findings showed the formation of a robust adsorption layer at the water−air interface and the possibility of transfer from the solution's surface onto a freshly cleaved mica plate using the Langmuir−Schaefer technique. Reliable measurement results were obtained for the dynamic surface dilational elasticity and dynamic surface tension. The analysis of the experimental findings, in combination with the relevant atomic force microscopy (AFM) images, revealed a non-homogeneous layer structure, consisting of interconnected surface microaggregates of fullerenol molecules, as presented in Figure 14. The surface aggregates are not adsorbed from the bulk phase. Rather, they form in the surface layer as a result of structural rearrangements of the adsorbed molecules. The slow fullerenol adsorption is not controlled by diffusion but by an electrostatic adsorption barrier.

Akentiev et al. [127] further investigated the surface properties of fullerenol solutions, examining the dynamic behavior of adsorption layers with a smaller number of hydroxyl groups (i.e., C60(OH)$_{20}$) with respect to the fullerenols in C60(OH)$_{30}$ [126]. Similar to the more hydrophilic fullerenol, C60(OH)$_{20}$ forms a rigid adsorption layer that can be easily transferred onto a solid surface using the Langmuir–Blodgett and Langmuir–Schaefer methods. In both cases, the surface properties of the solutions are sensitive to small mechanical surface perturbations. The adsorption layer of C60(OH)$_{20}$ is more fragile than that of C60(OH)$_{30}$. The experimental findings for C60(OH)$_{20}$ show that the time evolution of the surface tension and surface dilational elasticity exhibit diverse and peculiar features, resulting from different kinds of area perturbation (i.e., continuous or periodic oscillations, expansion or compression). Regarding C60(OH)$_{30}$ solutions, the adsorption kinetics are faster than for C60(OH)$_{20}$ solutions, presumably due to the lower charge of the molecules and lower adsorption barrier.

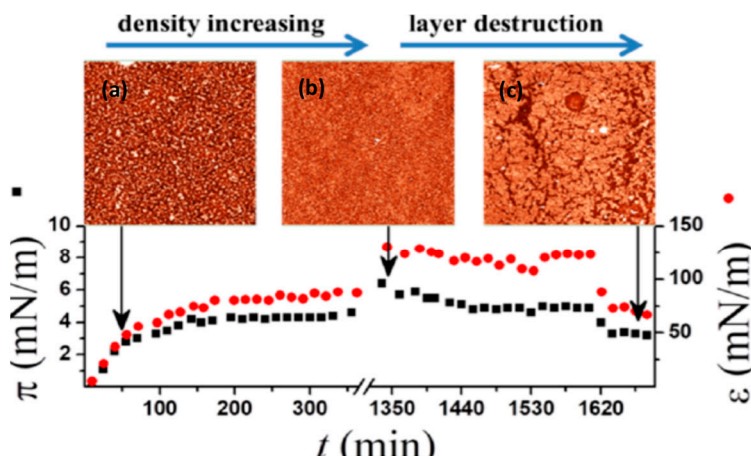

**Figure 14.** Upper side. AFM images of different samples of fullerenol C60(OH)x (x = 30 ± 2) film: (**a**) 70 min after the surface formation; (**b**) 16 h after the surface formation without oscillations; (**c**) after continuous oscillations for 75 min; lower side. The kinetic dependencies of the dynamic surface elasticity are shown with continuous barrier oscillations and various concentrations of fullerenol C60(OH)$_X$ (x = 30 ± 2), taken from [126].

Continuous oscillations of the surface area also lead to a decrease in the surface elasticity due to the partial degradation of the adsorption layer. During macroscopic observations, the adsorption layer appears homogeneous. However, the measurement values of the dynamic properties, supplemented with Brewster angle microscopy images, indicate that at the microscopic scale, the solution–air interface is non-homogeneous under specific conditions whereby the fullerenol molecules are arranged in multiple adsorption layers with separate surface aggregates.

Noskov et al. [128] characterized the structure of the complex spread-layer systems constituted by fullerene and its mixtures with the amphiphilic polymers poly(vinylpyrrolidone) (PVP) and poly(N-isopropylacrylamide) (PNIPAM), respectively. The interaction of fullerene with these polymers was investigated via measurements of dilational surface rheology and using optical techniques (i.e., ellipsometry, Brewster angle microscopy (BAM), and AFM).

The properties of these spread layers at the water–air interface indicated the strong adhesion of the layers to water, which can be explained by the hydroxylation of the fullerene molecules when they contact the water. The layers can sustain surface pressures of up to 70 mN/m. The dynamic dilational elasticity of the layers reaches values of approximately 370 mN/m at surface pressures close to 25 mN/m and then decreases gradually, giving rise to two local maxima. The ellipsometric measurements and BAM results show that the layers have variable thicknesses. According to the AFM data, the structure primarily comprises bilayers or trilayers at low surface pressures but can contain large aggregates of up to approximately 100 nm. The number of these tall aggregates increases as the surface pressure increases, but most of the surface aggregates have dimensions of approximately 40–60 nm in the x-y plane and approximately 20–40 nm in the z-direction. The local maximum of the surface elasticity can indicate the beginning of the layer's destruction. This conclusion was confirmed by optical micrographs that revealed the beginning of the formation of mesoscopic and macroscopic folds in the layer at surface pressures higher than 30 mN/m.

Protein–fullerene interactions were recently studied by Noskov et al. [7] by performing measurements of dilational surface elasticity as a function of surface pressure and surface age (together with AFM) on mixed layers of fullerene with bovine serum albumin (BSA) at the solution–air interface. The results from the dilational surface rheology and AFM measurements for C60/BSA layers at the water–air interface indicated significant interactions between the fullerene and protein molecules in the surface layer. The dependence of the module of the dynamic surface elasticity of the mixed C60/BSA spread layers on the surface pressure has two local maxima separated by a local minimum. The local minimum

corresponds to a partial displacement of the protein from the surface and divides the investigated range of surface pressures into two regions, with a prevailing influence of one of the components on the surface properties, as illustrated in Figure 15.

The protein influences the surface properties in the second region (at high surface pressures), while the fullerene can influence the surface properties in the first region. The AFM images show that both the C60 layers and mixed C60/BSA layers contain large fullerene aggregates with lengths in the z-directions of up to 100 nm. At the same time, the observed smaller aggregates in these two systems are different. The mixed layers contain some patches of the network of almost merged aggregates with a length in the z-direction of less than about 20 nm. This network presumably contains protein. The formation and the subsequent reorganization of the network lead to non-monotonic kinetic dependences of the dynamic surface elasticity module in the course of protein penetration into the fullerene layer. The formation of mixed protein/fullerene aggregates at the interface can presumably lead to a decrease in the fullerene's cytotoxicity due to the formation of the mixed protein/fullerene corona around the aggregates.

The reviewed scientific achievements [7,125–128] constitute a reliable set of measurement results that fruitfully serve as promising directions for further incoming studies. The reported interfacial dynamic properties together with the relevant applied methodologies and procedures will broaden the utilization of fullerene molecules in the near future (i.e., fullerenol solutions, fullerene–polymer and fullerene–protein mixed layers) in developing targeted pharmaceutical formulations for the improvement of special tumor therapies.

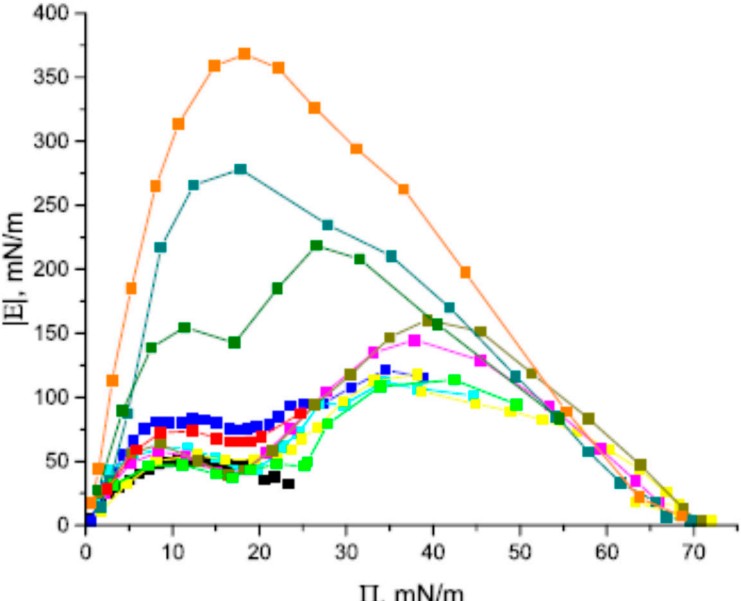

**Figure 15.** Module of dynamic surface elasticity of mixed C60/BSA, pure BSA (**black**), and pristine C60 (**orange**) spread layers as a function of the surface pressure at C60/BSA molar ratios of 88 (**red**), 117 (**green**), 153 (**blue, cyan**), 277 (**magenta**), 392 (**yellow, dark yellow**), 877 (**olive**), and 1660 (**dark cyan**); data taken from [7].

## 8. Summary and Conclusions

Medicine is a multidisciplinary scientific field and requires various types of input from other scientific branches, in particular from chemistry and physics. Surface science, combining physical and chemical principles, is, therefore, of great value for solving medical problems. Many recent developments in surface science have turned out to be suitable for medical diagnostics and the control of therapies for various diseases, as well as aiding in the development of drugs and their optimum delivery.

We have discussed some selected examples to demonstrate how experimental instrumentation, theoretical approaches, and model studies help to gain insight into the special

and complex situations doctors are often confronted with. The first example shows how measurements of the dynamic surface tension of human liquids can support the diagnostics of various diseases. After the first measurements by Künzel [10] of cerebrospinal liquor demonstrated impressively that different values for healthy and sick patients can be observed, systematic studies were performed using this methodology. Kazakov and co-workers summarized in [2] how the dynamic surface tensiometry of many human liquids gives invaluable inputs into diagnostics and therapy control for various medical branches. However, there is still a lot of unused potential because the rather conservative view in medicine prevents a broader application of these methodologies and does not allow to tensiometry to be applied as a cheap and efficient support tool.

An important second medical field deals with the formulation of artificial tears or synthetic lung surfactants. For both groups of diseases there is a lack of particular liquids that are produced normally by the healthy human body. To replace such very complex liquids, it is essential to understand which compounds are responsible for a required performance or function. For quite some time, systematic investigations of natural and synthetic tears have been under way. Already in 1996, for example, the stability of the tear film on the eyeball was systematically studied [129] as a vital prerequisite for the proper function of the eye. In the meantime, many new aspects have been addressed and will have to be investigated. There have also been historical studies on the functionality of pulmonary surfactants; for example, those performed by the pioneering teams of Possmayer [130], Schürch [131], and Neumann [132]. New experimental methodologies are being used to evaluate the administration of drugs to the lungs [133]. For the successful treatment of malfunctions of the lungs, most recently the work by the leading teams of Gil [134], Noskov [86], and Sosnowski [135] have been mentioned as being suitable for better formulating and evaluating the respective replacement liquids.

Most recently, an overview of the role of lung surfactants was published by Ravera et al. [81]. The various functionalities of the various liquids involved in the digestion process were discussed and the methods were analyzed to ensure they are suitable for systematic investigations in order to find optimum solutions for the detected problems.

One of the serious challenges in modern industrial countries is obesity. It represents a serious problem for the healthcare systems in many countries and it has turned out that surface science methods are quite successful in controlling energy intake in order to avoid overweight. New procedures are under way to reduce energy transfer to the body, despite the uptake of food in excess. During the coronavirus pandemic, it was even found in experiments with mice that the viral infection has an impact on obesity problems [136]. Using a particular methodology based on tensiometry, the entire digestion process can be successfully mimicked, as was demonstrated by Maldonado-Valderrama [105,106] using a multiliquid dosing system such as OCTOPUS combined with drop profile analysis tensiometry as a tool to mimic the digestion process, analogous to the system earlier used by Reis et al. [109].

A final example discussed here is the use of fullerenes for the diagnosis of cancer. In particular, the studies by Noskov and his team [7,126] on the interactions of different fullerenes with other types of surface-active molecules, such as polymers and proteins, provided much additional insight into the interfacial aspects of various cancer diseases.

The analyzed examples are of course only a selection, and a complete overview cannot be given in a single review article. We can conclude, however, that surface science provides very useful tools for the solution of biomedical problems. This situation, i.e., support given by surface science to various kinds of applied sciences and technologies, will surely further propagate. The observed great progress will further be supported by theoretical approaches, including thermodynamics, quantum chemical calculations, and molecular dynamics simulations. In order to get closer to reality, more complex mixed layers and further targets such as drug delivery will be in the schedule for future work.

**Author Contributions:** Conceptualization: R.M. and G.L.; writing—original draft preparation, review and editing: E.S., I.N., A.G.B., F.R., L.L., S.D., A.J., R.M. and G.L. All authors have read and agreed to the published version of the manuscript.

**Funding:** This research received no external funding.

**Data Availability Statement:** Not applicable.

**Conflicts of Interest:** The authors declare no conflict of interest.

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
