# Peer review of "Interfacial Dynamics of Adsorption Layers as Supports for Biomedical Research and Diagnostics"

_colloids, doi:10.3390/colloids6040081_

Round 1

Reviewer 1 Report

This paper reviews the application of surface tension and interfacial rheology characterization methods in a few biological and biomedical applications, including body fluids, artificial tear and lung surfactant, obesity, and cancer treatments. The paper reads well. I’d recommend a few minor clarifications to further improve the quality of this work. 

1.      I found the current title of the paper, “Interfacial dynamics of adsorption layers characterizing products for pharmaceutical formulation and medical therapy” very confusing. To be honest, I don’t know the meaning of this title.

2.      Line 388: The cause of RDS is not “impairment of its composition” but deficiency of lung surfactant due to prematurity.

3.      Line 426: a reference should be given to “constrained drop surfactometer”.

4.      Lines 485-489: for the discussion of surfactant therapy to treat COVID-19 patients, the authors may refer to a recent review article “The COVID-19 pandemic: a target for surfactant therapy?” by Veldhuizen et al.

5.      The sentence from line 51 to line 54 is too long and is hard to understand.

6.      Line 59: “corona pandemic” should be “COVID-19 pandemic”

7.      Line: 274: “… the second most frequent …” should be “ … the second most frequently used … ”

8.      Some figure captions contain copyright information (e.g. Fig. 11) but some others do not (e.g., Fig. 13). They may make the figure captions more uniform.

Reviewer 2 Report

The topic of the review is really interesting. However, the different parts of the review should be better harmonised since it could be a little dispersive. For instance, paragraph 2 is about analyses on biological liquids while paragraphs  3-7 are on in vitro investigations on lab-produced models. I suggest focusing better the review or connecting in a more organized manner all the paragraphs. In paragraph 2 the dynamic surface tension and dilational surface visco-elasticity does not emerge as a tool for diagnosis and therapy control in medicine. Paragraph 2.2 can be better described. Paragraph 2.3 is not about “various diseases” but about pregnacy and it is not clear to which “various diseases the title refers to. Figure 3 lacks numbers in the right Y-axis.

Round 2

Reviewer 2 Report

The authors have improved the manuscript and it is ready for publication